A comparison of vehicle emissions control strategies for smart cities

Tripp-Barba Carolina 1
Barbecho Pablo 2
Urquiza Luis 3
http://orcid.org/0000-0003-2048-9600 Aguilar-Calderón José Alfonso 1 ja.aguilar@uas.edu.mx
1 Facultad de Informática Mazatlán, Universidad Autónoma de Sinaloa , Mazatlán , México
2 Departamento de Ingeniería Eléctrica, Electrónica y Telecomunicaciones, Universidad de Cuenca , Cuenca, Azuay , Ecuador
3 Departamento de Electrónica, Telecomunicaciones y Redes de Información (DETRI), Escuela Politécnica Nacional , Quito , Ecuador
Ahmad Ayaz
Electronic publication date: 2023 Nov 6
Publication date: 2023
Volume: 9
Electronic Location ID: e1676
Received 2023 Jun 28; Accepted 2023 Oct 10
Copyright: © 2023 Tripp-Barba et al.
Copyright year: 2023
Copyright holder: Tripp-Barba et al.
License: This is an open access article distributed under the terms of the Creative Commons Attribution License, which permits unrestricted use, distribution, reproduction and adaptation in any medium and for any purpose provided that it is properly attributed. For attribution, the original author(s), title, publication source (PeerJ Computer Science) and either DOI or URL of the article must be cited.
License URL: https://creativecommons.org/licenses/by/4.0/

Keywords: VANET, Smart cities, Vehicle emissions control, Air pollution, Vehicular networks, Vehicular ad hoc networks

Funding: Universidad Autónoma de Sinaloa (UAS) PRO-A8-033 Escuela Politécnica Nacional PIGR-22-06 This work was supported by the PROFAPI (Programa de Fomento y Apoyo a Proyectos de Investigación) project: PRO-A8-033 from Universidad Autónoma de Sinaloa (UAS), México, and Escuela Politécnica Nacional through research project PIGR-22-06 “APlicaciones con Privacidad para Sistemas de TRAnsporte impulsados por DAtos (APPSTRADA)”. The funders had no role in study design, data collection and analysis, decision to publish, or preparation of the manuscript.

==============================
Many studies have shown that air quality in cities is affected due to emissions of carbon from vehicles. As a result, policymakers (e.g., municipalities) intensely search for new ways to reduce air pollution due to its relation to health diseases. With this concern, connected vehicle technologies can leverage alternative on-road emissions control policies. The present investigation studies the impact on air pollution by (i) updating vehicles’ routes to avoid pollution exposure (route choice policy), (ii) updating vehicles’ speed limits (speed control policy), and (iii) considering electric vehicles (EVs). Vehicles are informed in advance about route conditions (i.e., on-road emissions) using the vehicular network. We found that by updating vehicle routes, 7.43% less CO emissions are produced within the evaluated region. Also, we find no evidence of significant emissions reductions in the case of limiting vehicles’ speed. Lastly, with 30% of EV penetration, safe CO emissions levels are reached.

Introduction

As the world’s population continues to grow, urban areas are facing a range of environmental and health challenges. In 2018, more than half of the world’s population lived in urban areas, and this proportion is expected to increase to nearly 70% by 2050 (United Nations, 2018). The rapid pace of urbanization is putting pressure on cities to provide adequate infrastructure, including waste management, green spaces, and transportation.

One of the most significant challenges facing urban areas is air pollution. Air pollution is a major global environmental problem that is linked to a range of health issues, including respiratory disease, heart disease, and stroke (Balen et al., 2020). The World Health Organization (WHO) reports that nearly the entire world population breathes air that exceeds the recommended limits for pollutants, with exposure higher in low- and middle-income countries (WHO, 2022).

To address this problem, policymakers and researchers are exploring a range of solutions, including electric vehicles, renewable energy, and urban planning (Yadlapalli et al., 2022). Electric vehicles are up to four times more efficient than internal combustion engines and can be powered by locally generated renewable energy. By reducing greenhouse gas emissions and air pollution, electric vehicles have the potential to make a significant impact on urban air quality and public health (Barbecho Bautista et al., 2022; Sadeghian et al., 2022).

In addition to electric vehicles, other solutions are also being explored, including policies to promote renewable energy, public transportation, and green urban spaces. By working together to address the environmental and health challenges of urbanization, policymakers, researchers, and communities can create sustainable, healthy, and livable cities for all (Sanguesa et al., 2021).

The vehicular growth in Mazatlan is documented as of 2013 with the opening of the Mazatlan-Durango highway. The new and modern route shortened travel times from nearby states intending to grow as a tourist destination, but this implied other unforeseen aspects, such as traffic chaos. The National Institute of Statistics and Geography of Mexico (INEGI, 2020), indicates that there are 501,441 inhabitants in 2020. Meanwhile, that year, 222,022 motor vehicles circulated in the city, including vehicles, motorcycles, transport trucks, and public service. These data reveal that we live in a city where vehicles are practically the equivalent of half of its population. According to the public information, the other interesting fact is that in 2022, the number of vehicles was 262,444 (INEGI, 2021). This equals 20,211 new cars per year. If the trend continues, this 2023 Mazatlan will add around 20,000 new vehicles in circulation, going to 282,655 cars (Arias, 2023). This data doesn’t take into consideration the increase that the tourists add to this data.

According to Becerra Pérez & Ramos Álvarez (2020), agro-industrial activities, mining, and electricity generation are Sinaloa’s primary fixed sources of air pollution. In contrast, the sources of mobile pollution are motor vehicles, forest fires, and agricultural burning. The analysis presented by Becerra selected PM 2.5 as an indicator of air pollution. Typically, this pollutant has internal combustion engines (automobiles) as its primary source, industrial activities, electricity generation, mines, dust resuspension, agricultural burning, and some natural factors such as forest fires and volcanic emissions. The results show an average annual concentration of 22.20 ( μg/m3) for Mazatlan when the average data according to international and national standards are between 10 and 12 ( μg/m3). Due to these results, the authors conclude that between 144 and 171 out of 2,423 deaths due to these pollutants, could be avoided by reducing PM 2.5 concentrations.

With this background in mind, this work proposes an emissions control strategy using vehicular ad hoc networks (VANET) communication and the presence of electric vehicles. The proposal introduces constant vehicle emission measurement in the streets, intended to change the route when zones with high pollutant concentrations are found. The main objective is to evaluate the behavior of the route choice and speed-control policies established in the proposal, focusing on a zone of Mazatlan city (Mexico) as a scenario through simulations. In this case, the traffic flow in the area and the vehicle distribution were evaluated, and the CO emissions with the service were enabled and disabled—also, the impact of the vehicle’s speed in the results.

Related works

Over the years, different works have been interested in analyzing the impact of EV and its behavior in air pollution. That is the case of the following. Many existing works focus on a variety of related topics. For example, some researchers analyze the impact of the movement of vehicles on air pollution, as in the case of Llano & Orozco (2014) where the authors focus on answering the question of how to increase the energy efficiency of moving vehicles (i.e., reduce fuel consumption and GHG emissions, particularly CO2) depending on the route. In Doolan & Muntean (2017) proposes EcoTrec, a novel routing algorithm that balances vehicular traffic smartly, reducing travel time and gas emissions along the vehicle’s route, and the algorithm proposed in Padrón et al. (2021) where the main goal is to check the effect such factors would have on the overall traffic circulation and how pollution varies in nearby areas. Finally, in Souza et al. (2014), an intelligent traffic information system based on inter-vehicle communication is proposed to avoid vehicle traffic congestion. The main goal of the proposed approach is to decrease CO2 emissions, notifying drivers of existing accidents and, consequently, congested street segments through a cooperative multi-hop dissemination process to decide whether to change the route.

Also, many researchers was focusing in the potential of electric vehicles (EV) improvement the air quality, as the study made in Germany by Breuer et al. (2021). In Hsieh et al. (2022), a study is provided on EVs’ currently unclear environmental and health benefits, particularly in China. Also, in Poland, a method for scenario analysis applied to study the reduction of exhaust emissions by introducing electric vehicles in a selected city was presented in Jacyna et al. (2021). In Sharma & Chandel (2020), the authors analyze diverse pollutants such as PM 2.5, NO x, CO, and HC, and for CO2, comparing conventional automobiles and electric vehicles (EVs) used in transport in the Indian city. The research results show that not for all vehicle classes, EVs are the better alternative to mitigate emissions. Also, Wang et al. (2021), during the coronavirus disease 2019 (COVID-19) pandemic, utilized ground and satellite observations of air quality to evaluate predictions of a comprehensive chemical transport model and found that the substantial traffic reductions. The findings provide evidence of the potential environmental benefits of entirely switching to EVs.

Other works use the concepts of the Internet of Things (IoT) to offer alternatives, as in Kanpur Rani & Vallikanna (2020), which proposes a system to monitor air pollutants and generate advanced alerts by forecasting the pollution level in the city. In Samee, Jilani & Wahab (2019) and Gomathi et al. (2022) introduces the artificial neural network and deep learning protocols to predict air pollution; in Jiyal & Saini (2020), the analysis was made via Arduino, and Esther Pushpam, Kavitha & Karthik (2019) developed web-based application to achieve the monitoring.

The comparative analyses showed that introducing electric cars in cities significantly reduced the emission of harmful exhaust compounds and improved clean air in the city. However, the previous proposals analyzed the benefits and made monitoring, but it needs proposals related to real-time reaction when the air pollution results are negative. Considering the previous works, in this research, we focus on monitoring the air pollution in the city by the RSD (roadside units), considering the emissions of the vehicles passing in the area. That information is diffused through the VANET to change the route and distribute the traffic in real-time. Our proposal aims to control vehicle emissions and redistrict vehicles in alternative routes. The evaluation was made using accurate maps of Mazatlan, Mexico, evaluating the behavior if the model and analyzing trip length and duration.

System model

Maintaining air quality requires systematic monitoring, identifying the source of pollution, and adopting preventive actions. Nowadays, pollution coming from the transport sector can be collected using internet of things (IoT) devices (Kingsy Grace & Manju, 2019; Toma et al., 2019; Ullo & Sinha, 2020) or municipality-dedicated equipment (e.g., environment sensors) (Silva et al., 2019).

Jointly with communication technologies, innovative services appear in intelligent cities to control gas emissions from fossil-fuel vehicles (GAS vehicles) (Khan et al., 2019; Al-Bahrani et al., 2020). Novel technologies such as the vehicular network (e.g., C-V2X or DSRC vehicular communications -Dedicated Short Range Communications-), jointly with the city infrastructure (e.g., sensors, traffic control stations), enable new intelligent transport systems (ITS) developments in many areas: traffic control, dynamic route guidance, fuel consumption, and vehicle emissions control. Moreover, policymakers (e.g., municipality council) increasingly search for new ways to reduce air pollution.

In this work, we assess different policies intended to control emissions coming from the transport sector.

Figure 1 shows the system’s main components. First, an on-road measurement device allows the measurement of vehicle emission concentrations of CO emissions. This last is called a remote sensing device (RSD) and typically uses the principle that most gases will absorb light at particular wavelengths. It measures on-road emissions by absorbance of ultraviolet and infrared light across an open (optical) path using wavelength-specific detectors for different air pollutants. The RSD consists of an IR component for detecting CO, CO2, and NOx, together with a UV spectrometer for measuring emissions (Borken-Kleefeld & Dallmann, 2018; Huertas et al., 2020). Lastly, the amount of emissions measurements by an RSD is related to the space/volume surrounding the sensor’s location. Therefore, the measured pollution depends on the traffic flow in the road segment volume.

Figure 1 The general simulation scenario includes two types of vehicles: (i) fossil-fuel vehicles (GAS) and (ii) electric vehicles (EV).

In addition, the system considers on-road remote sensing devices (RSD) located on each road of interest (e.g., main street). Besides, city infrastructure comprises road-side units (RSU) intended to communicate route conditions (i.e., RSD measurements) to vehicles within its communication range.

Then, by taking advantage of vehicular communications, vehicles are informed in advance about route conditions (i.e., air pollution exposure); see Fig. 1. We assume vehicles are provisioned with communication capabilities (e.g., C-V2X, DSRC). At this point, two kinds of vehicles are included within the system model: (i) fossil-fuel vehicles and (ii) electric vehicles (EVs). The first case, named GAS, (see Fig. 1), refers to the typical case of vehicles powered by fossil fuels, such as gasoline (petrol) or diesel. The second case, named EV (see Fig. 1), refers to the electromobility case. Here, vehicles are provisioned with one or more electric motors for propulsion. Notice that we consider the full-electric type of vehicle as detailed in the following sections.

Finally, considering on-road remote sensing emissions, the system estimates short-time averaging thresholds to maintain safe air pollution exposure around the area of interest. Then, in case of air pollution exposure on a given road, the system will communicate to vehicles the policy to apply. Therefore, without loss of generality, we consider Dedicated Short Range Communications (DSRC) as the communication framework between vehicles and city infrastructure (i.e., roadside units), known as V2I communications.

Regarding the selected policy, in the case of GAS vehicles, two mechanisms to control emissions are evaluated: (i) route choice policy and (ii) speed-control policy. On the other hand, in the case of EVs, those referred to as the full-electric type of vehicles, non-emissions are generated. In this sense, we define an alternative case that considers the circulation of EVs. The following of this Section introduces emissions measurement and detail emission control mechanisms.

Vehicle emissions measurement

The measurement of exhaust emissions of vehicles includes typically particulate matter (mass or particle number -part per million (PPM)-) and gases, such as carbon monoxide (CO), but also NOx, hydrocarbons, methane, ammonia, etc., in a given air volume [g/m 3] with a time frame.

In this work, we consider CO emissions as the evaluation metric as it is directly related to human health concerns (interferes with oxygen delivery to the body’s organs). In this sense, CO emissions can be estimated based on the fuel transformation in energy using several factors such as the weight moved, distance, time, driver style, road conditions, and emissions factor of each specific vehicle (i.e., motor efficiency).

Besides, temperature, wind speed, wind direction, and relative humidity influence the concentration of gases in the area of influence [g/m 3]. Here, we focus on measurements performed at the road’s level (i.e., vehicle emissions). For this, we define the type of vehicle called GAS (Fossil-fuel Vehicle).

To assess different vehicle emissions control mechanisms, we consider boundaries imposed for noteworthy air pollutants related to health risks in The World Health Organization (WHO) air quality guidelines (WHO, 2021). The WHO has a long history of developing air quality guidelines for protecting human health. These guidelines are highly methodological and developed through a transparent, evidence-based decision-making process. As a result, they are widely used as a reference by policymakers worldwide in setting standards and objectives for air quality management.

Table 1 shows existing air quality guidelines for nitrogen dioxide, sulfur dioxide, and carbon monoxide with short averaging times. We use Table 1 to compare measured on-road pollutant concentrations from a health perspective to assess the effectiveness of assessed emissions control policies.

Table 1 WHO air quality guidelines (WHO, 2021) short time averaging pollutant thresholds.

Pollutant	Averaging time	Air quality guidelines	
NO 2 [ug/m 3]	1-h	200	
SO 2 [ug/m 3]	10-min	500	
CO [mg/m 3]	8-h	10	
	1-h	35	
	15-min	100	

Route choice policy

Typically, drivers’ route choices aim to minimize travel time. However, optimal path selection is also governed by many other criteria, such as avoiding route events (e.g., traffic jams, accidents), minimizing the number of traffic signals, and adding intermediate stops for charging the battery for electric vehicles.

Here, we propose a route selection scheme to preserve safe air quality in dense areas. The proposed policy is detailed in Algorithm 1. It considers two main actors: (i) the public infrastructure, in charge of updating air quality levels, in lines 1–9 in Algorithm 1, and (ii) private citizen vehicles, running throughout cities, in lines 10–20 in Algorithm 1.

Algorithm 1 Route choice policy

Data: R, a set of n road Ids considered within the emissions-controlled area. S, a sorted list of	
    roadIds according to their RSD roadId readings. V edges, a set of edges that comprises the	
    current vehicle’s route.	
Result: Vehicles select less polluted routes based on AQE alert messages.	
/* Public Infrastructure Premises.	
1 for eachroadId ∈ R do	
2   update RSD roadId reading	
3   compute avrg(RSD roadId);/* Updated every 15 minutes.	
4   update S list;/* Tuple (roadId, avrg(RSD roadId))	
5   if avrgRSDroadId>AQG then	
6     sort S list;/* S list is sorted in ascending order according to	
      average (RSD roadId) entries.	
7     generate AQE message;/* AQE messages include roadId, roadLoc,	
      and sorted S list.	
8     RSUs broadcast AQE messages;/* I2V communication.	
9   end	
10 end	
 /* Vehicles Premises.	
11 if AQE message received then	
12   if NodeCurrTime − AQETimestamp>TThreshold then	
13    from AQE message GET roadId, roadLoc, S list	
14    if AQE roadId ∈Vedges then	
15     recalculate vehicle route;/* Less polluted road is considered in	
        the first place, and so on; see Fig. 3.	
16    end	
17    if AQEhops≤Maxhops then	
18     forward AQE message/* V2V communication; see Fig. 2.	
19    end	
20   end	
21 end	

Figure 2 RSU broadcasts air quality exposure messages (AQE) to nearby vehicles within its communication range.

The receiving nodes (R1, R2, R3, R4) forward AQE messages in a multi-hop manner (receiving node R5).

First, as part of the public infrastructure, on-road remote sensing devices (RSD roadId) periodically update roads’ emissions; see line 2 in Algorithm 1. Then, roads’ CO emission readings are averaged following a short period (15 min) as refereed in WHO guides (WHO, 2021), see line 3. Note that this last step is needed to allow the comparison of measured values against AQG safety emission limits (CO emissions <100 mg/m3, see Table 1), in line 4. Pollutant measures are treated as public information (non-private user information is included) and handled by municipalities’ public infrastructure.

In case of any road sensor reading exceeds safety emission limits (i.e., RSD roadId>AQG in line 5), the city infrastructure (RSUs) generate air quality exposure (AQE) alert messages, see line 7 in Algorithm 1. The AQE alert messages include road information regarding that road showing non-safety emission levels. Road’s report includes identifier ( roadId) and location ( roadLoc). Also, a sorted list in ascending order containing RSD roadId readings is included (S list); see line 6 in Algorithm 1. Data included in AQE messages is later used (at vehicle premises) to select alternative vehicles’ routes intended to preserve air quality levels throughout the controlled emissions area.

At this point, AQE messages are broadcasted using RSUs located at the controlled area; see line 8 in Algorithm 1. The communication follows a multi-hop scheme; see Fig. 2. In a first instance, recipient nodes (i.e., vehicles) are those within the RSU’s transmission range (nodes R1, R2, R3, and R4 in Fig. 2). In a second instance, receiving vehicles process the AQE message (see line 11 in Algorithm 1). Notice none of the driver’s private information is shared with the public infrastructure.

The AQE message validity is verified by subtracting the message’s Timestamp (AQE Timestamp) from the vehicle’s current time (Node CurrTime). The result is later compared against the expiration time ( TThreshold) in line 12. In this manner, the current node avoids processing outdated AQE messages due to vehicular network delays (Shahwani et al., 2022). Then, the road’s information is recovered from the received message in line 13 in Algorithm 1. In case of the notified roadId (i.e., road exceeding safety emission levels) is considered within the current vehicle’s route (see line 14), vehicle’s driver is notified about it, an is intended to find an alternative path in line 15.

At this point, an alternative path for the vehicle is selected by jointly considering (i) the safety route regarding contamination and (ii) the closest route in terms of time. Figure 3 presents an example of the route selection approach. The R2 route is selected as it fulfills emission limits and shows the lowest trip time (see Table in Fig. 3).

Figure 3 V1 vehicle, recalculate its route taking into account ordered less polluted roads in S list and the trip time till vehicle’s destination.

R3 road (less polluted route) is located at the first place in the S list, while R1 road is considered the banned road. In this example, the R2 route is selected as it suggests safety air quality and the lowest trip time between alternative paths to the non-safety R1 route.

First, vehicles re-compute alternative paths in real-time (as soon as they receive and validate the AQG alert), considering current traffic conditions (i.e., trip times on edges). In case of route events (e.g., traffic jam, accident), the route’s trip time increases, making it less likely to be selected. In this sense, traffic is effectively distributed across alternative paths and avoids congested adjacent lanes. Lastly, alternative routes are computed using Dijkstra’s algorithm as detailed in Lopez et al. (2018).

Second, the received AQE alert message includes the S list (see line 13 in Algorithm 1). The S list consists of those less polluted roads intended to be selected as alternative paths.

Finally, recipient nodes (R1, R2, R3, R4 in the example of Fig. 2) forwards the received AQG message vehicle to vehicle ( R1→R5 in Fig. 2), in line 18. If the hop count exceeds two hops, the receiving node (R5 in Fig. 2) refrains in the forwarding. This way, after two communication hops, no AQG message is forwarded. The forwarding mechanism focuses on vehicles within the emissions-controlled area. Also, it reduces network overhead compared to simple flooding, seeking to avoid congestion.

We made the following assumptions: Vehicles are equipped with a global position system (GPS) as well as with an onboard unit (OBU), which enables vehicles to receive RSU alert messages.

Vehicles are aware of smart city services via road-side units (RSUs) deployed along the city.

Vehicle’s route is locally available (e.g., Google Maps, GPS). An alternative route (avoiding notified non-safety road) can be computed using an onboard system (e.g., GPS).

The same weather conditions are assumed during the whole experiment.

Speed-control policy

Authorities increasingly consider limiting vehicles’ speed to improve air quality in densely populated cities. However, the effect of restricting vehicles’ maximum speed on the local air quality is hard to predict, as it depends on drivers’ behavior as well as on route conditions (e.g., the number of traffic lights). In this sense, evaluating the effect of vehicles’ speed on air quality is crucial.

Implementing this kind of policy into a real-world environment implies complex logistics and could be dangerous for drivers. Here, we propose a Speed-control policy to provide a better understanding of this control mechanism’s effectiveness. The effect of implementing a variable road’s speed limit is analyzed using a microscopic approach by focusing on individual driver’s behavior (e.g., acceleration and deceleration).

The speed-control policy is detailed in Algorithm 2. As in Algorithm 1, the set of n roads included in the R list considered within the emissions-controlled area iterated to updated sensor readings (RSD roadId) in line 2. Later, sensor readings are averaged in line 3. In case a road’s emission overcomes AQG safety limits (see line 4), the AQE alert message is generated and broadcasted in lines 5 and 7, respectively. Here, the AQE alert message includes road information (roadId and roadLoc) and a variable road speed (VRS) factor. This last is computed according to the target speed to be evaluated ( TSpeed) concerning the maximum road speed given by the municipality council in line 5.

Algorithm 2 Speed-control policy.

Data: R, a set of n road Ids considered within the emissions-controlled area. VRS, variable road speed factor. RMS roadId, Road Maximum	
    Speed. TSroadId, target vehicle’s speed.	
Result: Vehicles adjust its current speed to the TSroadId.	
1 for each roadId ∈ R do	
2   update RSD roadId reading;	
3   compute avrg(RSD roadId);/* Updated every 15 minutes.	
4   if avrg RSD roadId> AQG then	
5     VRS=TSroadId/RMSroadId;	
6     generate AQE message;/* AQE messages include roadId, roadLoc,	
      and VRS factor.	
7     RSUs broadcast AQE messages;/* I2V communication.	
8   end	
9 end	
 /* Vehicles Premises.	
10 if AQE message received then	
11   if NodeCurrTime − AQETimestamp>TThreshold then	
12    from AQE message GET roadId, roadLoc, VRS factor;	
13     if CarroadId==AQEroadId then	
14       CarSpeed=VRS∗RMSroadId;/* Set current vehicle′s speed to the	
        target speed on roadId ( TSroadId).	
15     end	
16     if AQEhops≤Maxhops then	
17        forward AQE message/* V2V communication, see Fig. 2.	
18     end	
19   end	
20 end	

Lastly, at vehicle premises, in case an AQE message is received in line 10, the message is validated in line 11, and the road’s information, as well as the VRS factor, are recovered in line 12. At this point, the algorithm looks for the time the car reaches the recovered roadId in line 13. Here, the driver is intended to adjust the speed accordingly in line 14. Finally, as with the Route Choice Policy, the multi-hop communication scheme limit AQE alter message notifications; see lines 16–18 in Algorithm 2.

Simulation scenario

We consider a widely-known simulation framework to evaluate proposed vehicle emissions control strategies. First, the vehicular network is implemented using the open-source framework VEINS (Sommer, German & Dressler, 2011). The OMNeT++ framework (Vargas, 2018) acts as the network simulation platform. Then, the traffic mobility simulation is implemented in SUMO (Lopez et al., 2018). Besides, traffic elements (road maps, intersections, speed limit of streets, traffic lights) are imported directly from OpenStreetMaps (OSM) (Coast, 2018). In addition to road networks, we consider building structures (polygons) that may interfere with the signal between the sender (RSUs) and receiver (vehicles).

Figure 4A shows a medium-size dense urban scenario, with an area of 1,400 m × 1,400 m, from Mazatlan city, Mexico. It belongs to the Mexican state of Sinaloa, located on the coast of the Pacific Ocean. The town comprises tourist complexes along the coastal zone that runs over 17 km, making it one of the most extensive in the world. Mazatlan has grown in size and population due to the growth of infrastructure and tourism. This rapid growth and related civic activity have affected road traffic congestion, affecting air quality. Figure 4B shows the region of interest (ROI), Benito Juárez, that comprises the four most used avenues in Mazatlan: Av. Internacional, Av. Revolución, Av. Internacional, and Av. General Rafael Buelna.

Figure 4 Simulation scenario: (A) OSM Mazatlan city zone (Map data © 2023 Google); (B) Region of Interest (ROI) includes four main transited avenues in the area of Benito Juárez in the city of Mazatlán.

The mobility environment comprises fossil-fuel vehicles (GAS) and electric vehicles (EV), see Table 2. The first one includes an emissions device (Krajzewicz et al., 2015); the latter consists of a battery device (EVSpecifications, 2020). Regarding gas emissions, GAS vehicles emit a range of pollutants during fuel combustion (gasoline, diesel) based on the Handbook Emission Factors for Road Transport version 4 (HBEFA4). This provides different emission factors, either as weighted emission factors (per vehicle category), as emission factors per concept (e.g., conventional passenger cars, passenger cars with catalysts, diesel passenger cars), as emission factors per fuel type (gasoline, diesel) or as emission factors per sub-segment (Euro-4, Euro-5, Euro-6) common in urban areas. Exposure to CO emissions is associated with respiratory disease (including asthma), with symptoms such as coughing, wheezing, difficulty breathing, and more hospital admissions.

Table 2 Simulation settings for the scenario.

Parameter	Value	
Type of map	Urban (see Fig. 4)	
Simulation time	14 h	
Types of vehicles	GAS, EV	
Mobility model	Krauss model (Song et al., 2015)	
Emissions model	Emission factors per subsegment	
	HBEFA4-based (Krajzewicz et al., 2015)	
EV type	Kia Soul EV 2020 (EVSpecifications, 2020)	
EVs’ battery capacity	64 kWh	
EVs’ mass	1,830 kg	
EV energy model	SUMO model (Krajzewicz et al., 2015)	
Path loss model	Empirical IEEE 802.11p (Sommer et al., 2011)	
Transmission range	400 m in LoS	
PHY and MAC	IEEE 802.11p	
Nominal bandwidth	6 Mbps	
Road side units (RSUs)	4	
Beacon interval	1 s	
RSD (on-road remote sensing device)	4 (one per road)	

Besides, we consider vehicles are equipped with vehicular communication capabilities (e.g., WAVE/IEEE 802.11p, C-V2X). Without loss of generalization, we use the IEEE 802.11p standard on MAC and physical layers, see Table 2. On the one hand, the CO emissions notification service uses the channel SCH3 (174) to forward system information messages from RSUs to nearby vehicles. On the other hand, cars subscribe to the service via the CCH (178). RSUs are in charge of notifying vehicles regarding the air quality throughout the city. RSUs communicated CO road measurements in the city center (Av. Internacional, Av. Revolución, Av. Independencia, and Av. General Rafael Buelna). By continuously measuring CO emissions per road, the system can notify on-real time which roads overcome safety emissions levels. As soon as vehicles are notified of such an event, they recompute their route to avoid congested roads (i.e., roads with poor air quality). The main settings of the scenario are presented in Table 2.

Lastly, Fig. 5 shows a colormap of the mean vehicle density [ veh/km]. In red (10 veh/km on average) are the most congested roads during a working day. In dark blue, roads with smooth traffic flow are presented. Notice that we focus on the traffic flow in Benito Juárez. Here, Av. Independencia, Av. Revolución, Av. Internacional and Av. General Rafael Buelna shows a dense flow of traffic.

Figure 5 Mean vehicle’s density [veh/km] generated during a typical working day in the area of Benito Juárez.

In red, we can see the most congested roads. In dark blue, we can see roads with a smooth flow of vehicles.

Results

In this section, we assess different approaches to controlling vehicle emissions and the trends over time concerning traffic conditions (e.g., peak hours). During the simulation, GAS vehicles emit various pollutants due to fuel consumption. To assess the trained model, we measure the concentration of CO (short-time averaging) and fundamental traffic metrics, such as road occupancy, trip lengths, and duration. Figure 4A shows the area of impact, which includes the four main transited avenues in Benito Juárez in the city of Mazatlan. Besides, four CO sensors are located on roads, as depicted in Figure 4B. In the following, vehicle emissions control policies presented in System Model Section are evaluated. Lastly, an additional case is considered, including the presence of EVs.

Evaluation of route choice policy

Currently, no vehicle emissions control mechanism is used in the evaluated area of Benito Juárez (see Fig. 4B). In this sense, drivers select their route without previous knowledge of route conditions (e.g., air pollution exposure). This behavior increases traffic intensity because drivers choose main streets instead of secondary roads. Further, air pollution in road proximity is also affected.

To assess the impact of including the emission control scheme in this scenario, Fig. 6 shows the flow of vehicles per road in the area of Benito Juárez (see Fig. 11). We consider the flow of GAS vehicles (see Table 2). Besides, the considered emissions control scheme modifies vehicles’ route choice based on the WHO AQG pollution thresholds, as detailed in System Model Section.

Figure 6 Traffic flow in the simulation area of Mazatlán during a working day.

The scenario considers the GAS type of vehicles in Table 2. (A) Emissions control service disabled. (B) Emissions control service enabled. (C) The total flow of vehicles for both cases (A) and (B).

Figure 11 The flow of vehicles [veh/h] during working hours (6 am—8 pm) in the area of Benito Juarez, Mazatlán.

The type of vehicles includes electric vehicles (EVs) and fossil-fuel vehicles (GAS), described in Table 2. Also, two scenarios are evaluated: (i) without emission control (OFF) and (ii) emission control enabled (ON).

First, Fig. 6A refers to the current case in Mazatlán where no control scheme is applied. We can see a high traffic demand during the early hours in the morning (6–10 am) and at the end of the working day (5–7 pm) (i.e., peak hours). We can see most of the traffic runs on the Av. International and Av. Independencia. In the case of Av. Revolución and Av. General Rafael receives less traffic.

Second, Fig. 6B shows the case with the emissions control scheme. As soon as the system starts working at 6 am, we can see that traffic flow on the most congested street (Av. Internacional) is reduced. Here, vehicles select alternative routes following the route choice scheme detailed in System Model Section. This effect is noticeable during peak hours. In the morning, from 7 to 8 am, and in the afternoon, from 6 to 7 pm, half of the traffic on the Av. Internacional, shown in Fig. 6A, is distributed between surrounding streets, shown in Fig. 6B.

Besides, Fig. 6C shows the total traffic flow in the ROI (see 4 (b)) for both cases: (i) emission control service on (ON-GAS), and (ii) emission control service off (OFF-GAS). In the case of (i), a higher traffic flow circulates in the ROI. In the case of (ii), vehicles select alternative routes outside the ROI so that less traffic flow is shown in Fig. 6C.

Figure 7 shows the result in terms of CO emission considering a short time averaging period (15 min) for both cases: (i) no emission control is considered (i.e., current situation), and (ii) emissions control system is enabled. Besides, the WHO emissions threshold WHO (2021) is included as a point of reference (black dotted line, 100 [mg/m 3]). In the case of non-emissions control, in Fig. 7A, we can see that during peak hours in Av. Internacional, CO emissions exceed the CO WHO threshold. As shown in Fig. 6 most vehicles include this road in their routes so that most of the generated pollution is measured on this road. Here, people close to these roads are exposed to non-safety CO emissions.

Figure 7 CO emissions short time averaging (15 min) (A) Control services disabled (B) Control service enabled (C) Total CO in roads for both cases (ON/OFF).

In the other case, with the emissions control system enabled, in Fig. 7A, we can see that CO emissions are reduced for the Av. Internacional. Here, the route choice is handled considering information regarding route conditions (i.e., air pollution exposure) as detailed in Algorithm 1. We can notice that vehicles select alternative roads, reducing congestion and pollution on main streets. Particularly, cars choose the closest and less polluted roads. With the emissions control system enabled, alternative streets are selected (Av. General Rafael, Av. Revolución), increasing the flow of vehicles and, therefore, emissions. Furthermore, in Fig. 7B, we can see CO emissions increases for Av. Revolución, Av. Independencia and Av. General Rafael, due to the route choice navigation scheme. Here, taking advantage of the vehicular network, vehicles can receive information regarding route conditions. Vehicles’ onboard units select the optimal route jointly, considering vehicle density, travel time, and CO measurements on each road. In this case, it is clear that CO concentrations are substantially below the short-term WHO guidelines and would not be considered at a level that could have significant health impacts.

Even though the control scheme distributes vehicles throughout downtown roads maintaining CO below current WHO health-based guideline values (<100 mg/m3), in Fig. 7B, from 6 to 8 am, we can see that CO emissions are above the recommendation (115 mg/m3). This is due to the pollution’s time averaging period, meaning that vehicles are informed about route conditions following that interval. Lastly, in Fig. 7C, we can see that the control scheme reduces CO emission coming from GAS vehicles during the whole working day in the area by 7.43%.

In Fig. 8, we analyze the impact of using the route choice mechanism considering fundamental traffic metrics. First, we assess the mean trip length and trip duration. Then we evaluate the mean vehicle’s time loss. Time loss refers to vehicles driving at a lower speed than the maximum allowed speed on that road due to traffic lights or events on the route.

Figure 8 The flow of vehicles [veh/h] during working hours (6 am—8 pm) in the area of Benito Juarez, Mazatlán.

The type of vehicles includes electric vehicles (EVs) and fossil-fuel vehicles (GAS), described in Table 2. Also, two scenarios are evaluated: (i) without emission control (OFF) and (ii) emission control enabled (ON).

In contrast to what we can expect for the case of applying the route choice scheme, which modifies the original route to an alternative, less polluted path, it presents a non-representative difference in terms of vehicles’ route length when comparing against not using emissions control method, see Fig. 8. This is because the route choice scheme looks for the closest less polluted route in order to select an alternative path for vehicles (see System Model Section). Despite the obtained trip length distance, in Fig. 8, we can see that with the route choice scheme, vehicles require 21% more time to reach their destination. Besides, vehicles lose more time on the route due to secondary streets being selected and a higher number of traffic signals on the route.

Evaluation of speed-control policy

In this section, we evaluate the speed-control scheme detailed in Section 21. Here, we aim to control CO emissions to safe levels by controlling the vehicle’s speed. This scheme considers the presence of GAS vehicles (see Table 2).

In Fig. 9, we consider both cases: (i) no control scheme (current situation in Mazatlán), and (ii) speed-control scheme by considering speed limit reductions going from 55 to 25 km/h. In the first case, no control mechanism is applied (red dotted line in Fig. 9). Here, vehicles circulate along streets without restrictions other than the speed limit. In Fig. 9, we can see emissions overcome AQG recommendation (black dotted line), so that requires a control mechanism. In the second case, the maximum vehicle speed is restricted. In Fig. 9, vehicles running at low speeds (25 km/h) reach the highest CO emissions. Compared to the no emissions control case (purple column OFF), 6.32% more emissions are produced. Although, by considering 35–55 km/h, we can see in Fig. 9 that emissions are reduced in near to 10% for all the cases (i.e., 35, 45, 55 km/h) compared to the case of no emissions control.

Figure 9 CO emissions short time averaging (15 min) in Av. Internacional (see Fig. 4).

The speed-control emissions scheme is evaluated considering vehicles’ speed in a range of 25—55 km/h. The current case in Mazatlán, where no control mechanism is used, is the OFF case (red dotted line). The AQG limit references certain CO emission levels (see Table 1).

Even-though, emissions are reduced, non of the speed limit cases effectively reach CO emissions below the AQG threshold during peak hours. In contrast to the analysis presented in Folgerø, Harding & Westby (2020), by considering speed limit reductions, we see that this mechanism can contribute as an environment vehicles emissions control policy in case of low traffic flow.

In Fig. 10, we show the mean CO emissions generated on Av. Internacional. Here, we can see that the same behavior presented in Fig. 9 is repeated by considering the whole working day. With the vehicle’s speed limit of 25 km/h, which represents half of the current maximum vehicle speed in Mazatlán, the highest emissions are presented. Then, by considering cases of 25, 35, 45, 55 km/h, CO emissions are reduced compared to the case of non-emissions control. Notice that, Mazatlán municipality set the maximum road speed to 50 km/h so that the last case of a maximum vehicle speed of 55 km/h is considered just for test purposes.

Figure 10 Mean CO emissions [mg/m 3] generated on Av. Internacional during a working day.

The speed-control scheme is assessed considering different speed limits (25—55 km/h). The case where no control mechanism is applied is shown in the purple column (OFF).

Evaluation of alternative mobility case (electric vehicles)

Among many innovative solutions, electric vehicles (EVs) are considered the most promising alternative to reduce gas emissions in the transportation sector, although the decisive shift of drivers to use EVs has been delayed mainly due to low penetration of charging infrastructure (i.e., charging stations) throughout cities (Mukherjee & Ryan, 2020). In this sense, we consider the coexistence of GAS and EV vehicles in the scenario. This scheme considers the presence of GAS and EVs circulating in the city. Therefore, we set the total EV amount to 30% of GAS vehicles. Besides, EVs are uniformly distributed within each simulation hour, considering the entire traffic flow [veh/h].

Figure 11 shows the flow of vehicles [veh/h] for both cases when the vehicle’s emissions control scheme is active (ON) and when it is not (OFF). In the case of EVs, we can see the same flow of vehicles circulating through downtown roads in both scenarios. However, in the other case (GAS vehicles), we can see a higher flow of vehicles circulating through downtown roads in case of the control scheme is disabled. This is because vehicles are banned from roads with high gas emissions and therefore select other routes outside the ROI when the control scheme is enabled.

Figure 12 shows averaging CO emissions for this scenario. Notice that the control scheme considers the final target of 100 mg/m 3 evaluating a short time averaging (15 min) according to WHO AQG (WHO, 2021). We can see that the Av. International handles most of the traffic in Benito Juarez.

Figure 12 CO emissions short time averaging (15 min) (A) Control services disabled (B) Control service enabled (C) Total CO in roads for both cases (ON/OFF).

In the case of no control scheme shown in Fig. 12A, at the first hours (6–10 am), we observe non-safety CO measures >100 mg/m 3 according to WHO AQG. In contrast, alternative routes, including Av. Revolución, Independencia, and General Rafael, show safe CO emissions <100 mg/m 3 during the whole day (see Fig. 12A). When considering the control emissions scheme in Fig. 12B, we see GAS vehicles change their original route to the Av. General Rafael, originally the less polluted road and the closest one. Figure 12B shows that the scheme distributes traffic between possible routes to reduce CO emissions during high-traffic hours. Besides, Fig. 12C shows the total CO emissions produced by GAS vehicles throughout all routes. We can see that with the service active (ON-GAS), higher levels of CO are produced in case of high traffic demand.

Figure 13 shows a colormap plot of mean CO emissions over roads. In case no service is enabled, we can see that most pollution is produced over the Av. International in brown color. In case the control service is enabled, we can see pollution is reduced on that road shown in yellow-green color, according to results shown in Fig. 12B.

Figure 13 Mean CO emissions mg/m 3 over roads maps (A) Control services disabled (B) Control service enabled.

Conclusions

Policymakers continuously search for new ways to reduce air pollution in dense urban areas. In this work, we evaluate the behavior of two policies intended to control on-roads emissions.

First, the route choice policy considers alternative routes with low CO emissions. Results show that this policy reduces CO emissions by 7.43% throughout the service area. Vehicles also require 21% more time to reach their destination. This confirms the effectiveness of the proposal, where it is possible to observe the distribution of the vehicle density in the evaluated streets, selecting alternative routes when necessary. Even with the time of trajectory increases, pollution reduction is positively considerable.

Then, we evaluate the speed-control policy, which tries to reduce emissions by restricting maximum vehicles’ speed. Results show that considering low speed (set 25 km/h), emissions increase; while with speed between 35–55 km/h, emissions are reduced. This shows that a regular speed contributes to minimizing the emissions.

Finally, we consider an alternative case based on adopting electric vehicles. We obtain safe CO emission levels when the EV’s penetration reaches 30% of the total traffic flow. This confirms the premise that electric vehicles can significantly reduce air pollution and greenhouse gas emissions. Using electric vehicles instead of vehicles that run on fossil fuels can reduce carbon footprints and air pollution, which benefits human health and the environment.

There is a fact that electric vehicle production increases productivity and job creation with a relatively small impact on the environment. However, there still needs to be more public policies by local governments to carry this out. Establishing campaigns promoting electric vehicle adoption and formation in primary education is necessary. This will produce capabilities, skills, knowledge, and facilities to successfully introduce electric vehicles as a helpful service or product to support national low-carbon goals. This will support the national policy established in the National Strategic Programs for Climate Change by the Mexican federal government because the environmental benefits like cleaner air, less pollution, climate change mitigation, and better health are actual. Until now, there have been no legacy validation or proposals regarding establishing electric vehicle-related environmental law, some tax credit planning for acquisition to make these vehicles affordable in the current market, laws for pollution control through and vehicular checks, laws for speed control in touristic and center areas of the city. In Mexico, it is a complicated task since each local government (city) has its on-traffic law system. So, there is no single system standardized for the entire country. Moreover, regulation in the local construction policies is mandatory since the infrastructural change must accelerate electric vehicle adoption.

Implementing policies and programs that encourage the use of electric vehicles is essential to accelerate their adoption. Some of these policies include fiscal and financial incentives, charging infrastructure, and the promotion of research and development of battery and electric motor technologies. However, there are still challenges in the mass adoption of electric vehicles, such as the need for charging infrastructure in some places, the higher initial costs, and the limited range of vehicles. These challenges must be addressed to make electric vehicles more accessible and affordable for the general public. In summary, we showed the efficiency of the approaches to controlling vehicle emissions developed in the present research, possibly reducing the concentration of pollution. We also present an evaluation that shows that electric vehicles are a promising solution to reduce pollution and greenhouse gas emissions.

As a future work identified in this research, it is essential to conduct a detailed analysis of the behavior of the proposals in this work using other pollutants, such as nitrogen oxides (NOx) and particulate matter (PM), that impact air quality and health and an exhaustive study of combinations of vehicle speeds and the effect of weather conditions.

The authors would like to thank the anonymous reviewers for the constructive comments, which helped improve this article. Special thanks to Universidad Autonoma de Sinaloa (UAS, Mexico) and to the Cuerpo Academico Tecnologia Educativa I+D+I (UAS-CA-303) for their guidance to the members of the group.

Additional Information and Declarations

Competing Interests

Author Contributions

Data Availability

The authors declare that they have no competing interests.

Carolina Tripp-Barba conceived and designed the experiments, performed the experiments, analyzed the data, performed the computation work, prepared figures and/or tables, authored or reviewed drafts of the article, and approved the final draft.

Pablo Barbecho conceived and designed the experiments, performed the computation work, authored or reviewed drafts of the article, and approved the final draft.

Luis Urquiza performed the experiments, authored or reviewed drafts of the article, and approved the final draft.

José Alfonso Aguilar-Calderón analyzed the data, performed the computation work, prepared figures and/or tables, authored or reviewed drafts of the article, and approved the final draft.

The following information was supplied regarding data availability:

The simulation code is available at Figshare: Tripp-Barba, Carolina; Urquiza-Aguiar, Luis Felipe; Aguilar-Calderón, Jose Alfonso (2023). Simulation code for comparison of vehicle emissions control strategies for smart cities. figshare. Software. https://doi.org/10.6084/m9.figshare.23581968.v1.

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
