# Peer review of "A comparison of vehicle emissions control strategies for smart cities"

_PeerJ Computer Science, doi:10.7717/peerj-cs.1676_

## Round 0.1 · original submission · Major Revisions

All the comments of the three reviewers should be addressed. Most importantly, a comparison with existing works is needed.

Reviewer 1 ·

Basic reporting

However, some Spelling and grammatical errors are found in some places:
1. Lines 136-142: The explanation is unclear, and there is also a repetition of a sentence.
2. Lines 381-382: Unclear
3. In line 364, There is a reference to a figure 11c instead of 10c
4. Line 99: "in" intelligent cities.
5. Additionally, Algorithm 1 and Algorithm 2 need proper formatting, and certain punctuation marks need correction.

Experimental design

The algorithm for the "route choice policy" (page6) needs to be expanded to provide a comprehensive explanation of how it effectively distributes traffic across various roads, mitigating congestion in adjacent lanes. It would be valuable to outline whether there exists a predefined distance threshold or time duration, after which no recommendations are suggested.

In Algorithm 1, could you elaborate on the process by which "tcurrent" is updated? Is this a 1minute incremental measurement over a rolling 15-minute period? For instance, is it calculated at 10:01 for the 15 minutes and then again at 10:02 for the subsequent 15 minutes? Kindly provide clarification.

Validity of the findings

Assessment of Speed-Control Policy: Please provide explicit clarification of any underlying assumptions. Whether the control and test groups subjected to trials on identical roads under similar weather conditions? Were there any events taken into account during testing?

Provide clarification regarding the method employed to acquire this data, along with details concerning the volume of data.

Additional comments

The problem is well formulated and the proposed solutions are well defined. However, I request the authors to provide more detailed information to make the findings more reliable.

Cite this review as

Reviewer 2 ·

Basic reporting

The paper addresses an important and timely issue of environmental and health challenges faced by urban areas due to population growth and urbanization. The focus on air pollution, a major problem in urban settings, is commendable. The proposal of using vehicular ad hoc networks and electric vehicles to mitigate emissions is an interesting approach worth investigating. However, the paper requires significant improvements to strengthen its arguments and address potential weaknesses.

Experimental design

Strengths:

Relevance: The paper tackles a critical problem faced by urban areas, making it relevant and potentially impactful for policymakers and researchers.

Comprehensive Measurement: The focus on carbon monoxide (CO) emissions and consideration of various influencing factors provides a comprehensive assessment of the emissions control strategies.

Simulation Framework: The use of a simulation scenario based on a real-world urban area allows for practical evaluation and analysis of the proposed strategies

Validity of the findings

Lack of Comparative Analysis: The paper lacks a comprehensive comparison with existing emissions control approaches. A comparative analysis would have added depth to the research and highlighted the novelty of the proposed strategies.

Limited Environmental Parameters: While the paper concentrates on CO emissions, other pollutants such as nitrogen oxides (NOx) and particulate matter (PM) also significantly impact air quality and health. The exclusion of these pollutants weakens the study's overall effectiveness in addressing urban air pollution comprehensively.

Uncertainty in Speed-Control Policy: The paper acknowledges the uncertainty in the impact of speed restrictions on local air quality due to driver behavior. This uncertainty undermines the credibility of the speed-control policy's effectiveness.

Areas for Improvement:

Robustness of Simulation: The paper should provide a thorough validation of the simulation framework against real-world data to ensure the accuracy and reliability of the results.

Sensitivity Analysis: Conducting a sensitivity analysis on key parameters such as electric vehicle adoption rate and route choice criteria would strengthen the paper's conclusions and recommendations.

Policy Implications: The paper should provide a more in-depth analysis of the economic, social, and environmental implications of implementing the proposed strategies, including costs and potential barriers to implementation.

Additional comments

Persuasive Argument for Improvement:
While the paper presents a relevant topic and some commendable aspects in the evaluation of emissions control strategies, it falls short in several crucial areas. A comprehensive comparison with existing approaches is essential to establish the superiority of the proposed strategies. Additionally, the exclusion of other pollutants such as NOx and PM weakens the study's ability to present a holistic solution to urban air pollution.

The uncertainty surrounding the speed-control policy raises questions about its practicality and effectiveness. More robust evidence and analysis are required to justify the potential impact of this policy on air quality improvement.

To enhance the credibility of the research, the authors should conduct a thorough validation of the simulation framework using real-world data. Furthermore, performing sensitivity analysis would strengthen the paper's conclusions and contribute to a more nuanced understanding of the strategies' potential.

Lastly, the paper should delve into the policy implications of implementing the proposed strategies. Policymakers need to understand the economic, social, and environmental costs and benefits of such measures to make informed decisions and create effective policies.

Cite this review as

Reviewer 3 ·

Basic reporting

The work makes it possible to investigate how certain changes in the flow and vehicle fleet impact air quality. I recommend its publication with minor revisions:

I suggest the author write a paragraph about the city of Mazatlán. What are the levels of pollution? Impact of air quality on the population? Previous air quality studies in the city?

The Handbook Emission Factors for Road Transport (HBEFA) are suitable for vehicle technologies circulating in Mexico?

Did the author consider evaluating the daytime cycle of vehicle circulation? I didn't see any graphs with peak traffic hours in the assessed area.

I did not understand the working principle of RSD and RSU. Please, rewrite!What is observational data and what is modeling?

Experimental design

I did not understand the working principle of RSD and RSU. Please, rewrite!What is observational data and what is modeling?

Validity of the findings

Smart cities are an important concept in public policies to reduce pollution. The autor maybe can explore that few modifications on urban mobility can reduce air pollution (carbon footprint) but policies for the use of electric vehicles are expensive and changes in mobility patterns (speed, for example) require a complex analysis of traffic, especially in center areas.

Cite this review as

---

## Round 0.2 · accepted · Accept

The authors have addressed the reviewers' comments and the authors have agreed with the revisions.

Reviewer 2 ·

Basic reporting

Good.

Experimental design

The authors have validated the study by using a simulator.

Validity of the findings

The results correspond to the findings of the paper.

Additional comments

The authors have successfully addressed my comments from the previous round. This has greatly improved the quality of the paper.

Cite this review as

Reviewer 3 ·

Basic reporting

All previous suggestions were accepted. The article is clearer and has a relevant scientific impact. I recommend your publication.

Experimental design

The use of a simulation based on a real-world urban area can improve the practical evaluation and analysis of the Smart Cities strategies. Congratulations!

Validity of the findings

As previously stated, few modifications on urban mobility can reduce air pollution, the authors compare policies for using electric vehicles, the increase of these vehicles, and the pollution increase in tourist areas and center areas, focusing on speed control laws and air pollution.

Cite this review as